# Correlation between Baseline Conventional Ultrasounds, Shear-Wave Elastography Indicators, and Neoadjuvant Therapy Efficacy in Triple-Negative Breast Cancer

**DOI:** 10.3390/diagnostics13203178

**Published:** 2023-10-11

**Authors:** Siyu Wang, Zihan Lan, Xue Wan, Jingyan Liu, Wen Wen, Yulan Peng

**Affiliations:** Department of Medical Ultrasound, West China Hospital, Sichuan University, Wai Nan Guo Xue Xiang 37, Chengdu 610041, China; wsy1051@163.com (S.W.); zinn_zihanlan@163.com (Z.L.); wanxueyouki@163.com (X.W.); jingyan.liu@foxmail.com (J.L.); wendywen072@163.com (W.W.)

**Keywords:** triple-negative breast cancer, conventional ultrasound, shear-wave elastography, neoadjuvant therapy efficacy

## Abstract

In patients with triple-negative breast cancer (TNBC)—the subtype with the poorest prognosis among breast cancers—it is crucial to assess the response to the currently widely employed neoadjuvant treatment (NAT) approaches. This study investigates the correlation between baseline conventional ultrasound (US) and shear-wave elastography (SWE) indicators and the pathological response of TNBC following NAT, with a specific focus on assessing predictive capability in the baseline state. This retrospective analysis was conducted by extracting baseline US features and SWE parameters, categorizing patients based on postoperative pathological grading. A univariate analysis was employed to determine the relationship between ultrasound indicators and pathological reactions. Additionally, we employed a receiver operating characteristic (ROC) curve analysis and multivariate logistic regression methods to evaluate the predictive potential of the baseline US indicators. This study comprised 106 TNBC patients, with 30 (28.30%) in a nonmajor histological response (NMHR) group and 76 (71.70%) in a major histological response (MHR) group. Following the univariate analysis, we found that T staging, dmax values, volumes, margin changes, skin alterations (i.e., thickening and invasion), retromammary space invasions, and supraclavicular lymph node abnormalities were significantly associated with pathological efficacy (*p* < 0.05). Combining clinical information with either US or SWE independently yielded baseline predictive abilities, with AUCs of 0.816 and 0.734, respectively. Notably, the combined model demonstrated an improved AUC of 0.827, with an accuracy of 76.41%, a sensitivity of 90.47%, a specificity of 55.81%, and statistical significance (*p* < 0.01). The baseline US and SWE indicators for TNBC exhibited a strong relationship with NAT response, offering predictive insights before treatment initiation, to a considerable extent.

## 1. Introduction

The incidence of breast cancer (BC) is on the rise, and within this category, triple-negative breast cancer (TNBC) accounts for 10–20% of all cases. TNBC is characterized by the absence of estrogen receptor (ER), progesterone receptor (PR), and human epidermal growth factor receptor 2 (HER2) expression [1], making it clinically known for its aggressive behavior, high recurrence rates, and propensity for metastasis. Due to the heterogeneous nature of TNBC and the limited effectiveness of treatments beyond chemotherapy, it stands as the subtype with the bleakest prognosis [2,3]. Neoadjuvant therapy (NAT) has emerged as a common strategy in BC treatment, as it can reduce tumor size, downgrade the cancer stage, and increase the likelihood of surgical resection and breast-conserving surgery. The Chinese Breast Cancer Treatment Guidelines [4] recommend NAT, particularly for HER2-positive (HER2+) BC or TNBC with specific tumor loads, to gain in vivo drug sensitivity information, optimize post-surgery drug selection, and enhance overall prognoses.

Achieving a pathologic complete response (pCR) following NAT is now considered a surrogate endpoint for long-term survival and disease-free periods [5]. Especially in high-risk BC subtypes, such as TNBC, the incidence of pCR in anthracycline and taxane NAT regimens ranges from 18% to 60% and varies depending on the TNBC subtype [6,7]. Consequently, studies have aimed to predict NAT responses early on and to develop noninvasive imaging markers capable of identifying potential pCR patients promptly, allowing for treatment plan adjustments.

An ultrasound (US) examination, as one of the commonly used breast imaging methods, is more suitable for Asian women’s dense breasts and is more economical and convenient in the medical environment in China. Research has proven the advantages of ultrasound in differentiating between benign and malignant breast tumors [8], and past studies have proven its value in BC treatment processes [9]. Additionally, the obvious interstitial reaction in and around breast cancer focus areas leads to the proliferation of fibrous tissue, resulting in increased stiffness [10,11]. Shear-wave elastography (SWE) is a noninvasive ultrasound imaging technique that measures the stiffness characteristics of breast tissue [12,13]. It provides quantitative elastic parameters by utilizing the acoustic radiation force caused by the ultrasonic push pulse generated by the transducer, and it is a quantitative representation of palpation by clinical doctors. Because it is more objective and exhibits high reproducibility [14] and high specificity for the diagnosis of BC, it has become one of the preferred examination techniques before invasive breast biopsy [15]. Past researchers believed that continuous decreases in mass stiffness during treatment processes can be good markers for prediction [16,17,18], but no specific analyses have been conducted on BC subtypes. Moreover, a surgical plan should be developed based on baseline characteristics, tumor response, and patient willingness after NAT [19]. Therefore, we aimed to advance the prediction timing for TNBC cases with poor prognoses, improve the ability to predict NAT responses at baseline, and assist patients in their treatment selection.

Our study sought to investigate the correlation between baseline US features and SWE parameters in TNBC patients exhibiting post-NAT pathological responses, as well as the predictive potential of combined imaging modes for NAT efficacy.

## 2. Materials and Methods

### 2.1. Study Population

The records of BC patients treated at the West China Hospital of Sichuan University were retrospectively analyzed from April 2020 to April 2022. The inclusion criteria were (1) the presence of biopsy-proven invasive TNBC [20] with no distant metastasis; (2) available baseline conventional US and SWE examination data; (3) patients scheduled for the implementation of standard TNBC NAT treatment; and (4) patients who underwent surgery after NAT was completed. The exclusion criteria were (1) multifocal or bilateral BC; (2) a history of BC or other treatments; (3) patients who were pregnant or lactating; (4) a baseline US examination at >30 days before NAT or unsatisfied image quality; (5) patients who had not yet completed NAT; and (6) remote metastases during NAT. Disease staging was performed according to the eighth edition of the American Joint Committee on Cancer (AJCC) TNM classification [21]. The study participants received comprehensive NAT treatments in accordance with established protocols based on anthracyclines and taxanes, following the guidelines of the National Comprehensive Cancer Network (NCCN) [22]. A flowchart depicting the selection criteria is presented in Figure 1.

### 2.2. Conventional US Features

The baseline ultrasound indicators of all masses were obtained within 30 days before NAT using a Siemens OXANA2 ABVS ultrasonic device (Siemens Healthineers, Munich, Germany). Image storage was completed by sonographers with over 5 years of experience in breast US diagnosis and who have undergone SWE technical training. Two of them jointly recorded the indicators. The reference criteria were as follows:

Conventional US: The section with the maximum diameter, the section perpendicular to the section with the maximum diameter, and all sections containing tumor features should be stored in the retrospective image. The characteristic manifestations were recorded based on ACR BI-RADS (Atlas, Breast Imaging Reporting and Data System) standards [23], including size, echo pattern, margins, orientation, calcifications, posterior acoustic effect, peripheral tissue involvement, and invasion layers. Furthermore, lymph node abnormalities were recorded at each level. Moreover, the Adler classification of CDFI (color Doppler flow imaging) was used to evaluate blood flow. Figure 2a–d show some necessary parts of the sections in the stored images.

SWE: The imaging conditions in grayscale images should be optimized to obtain SWE images based on WFUMB guidelines [24]; the sampling frame should cover the grayscale image of the entire lesion as much as possible. The patients were asked to hold their breath, and images were frozen and captured within 3–4 s after the sampling frame stabilized. The stiffness range from soft to hard was displayed in blue to red, 0–10 m/s. The region of interest (ROI) was set to 2 × 2 mm and placed in the position with the highest tumor stiffness. Five points of ROI in the hard part (red area) were selected to obtain different elasticity values (Figure 2e,f), and the machine automatically generated their maximum elasticity (Emax), minimum (Emin), median (Emedian), mean (Emean), standard deviation (Estd), and interquartile range (EIQR) for recording.

### 2.3. Outcome Grouping

Assessments of response to neoadjuvant therapy (NAT) utilized the Miller–Payne Grading (MPG) system [25], which relies on changes in cancer cellularity. This involved comparing the number of cancer cells in core biopsy samples before treatment with that in the resected tumor post-treatment:

Grade 1: No change, or some alteration to individual malignant cells, but no reduction in overall cellularity.

Grade 2: A minor loss of tumor cells, but overall cellularity still high (up to 30% loss).

Grade 3: An estimated reduction in tumor cells between 30% and 90%.

Grade 4: A marked disappearance of tumor cells such that only small clusters or widely dispersed individual cells remain (more than 90% loss of tumor cells).

Grade 5: No malignant cells identifiable in sections from the site of the tumor; only vascular fibroelastotic stroma remains often containing macrophages. However, ductal carcinoma in situ (DCIS) may be present.

In this study, participants with Grades 1–3 were assigned to the nonmajor histological response (NMHR) group, while participants with Grades 4 and 5 were assigned to the major histological response (MHR) group.

### 2.4. Statistics

A data analysis was conducted using SPSS version 26.0 (SPSS Inc., IBM Corp., Armonk, NY, USA). Qualitative image features were assessed using Pearson’s chi-squared test or Fisher’s exact test, while quantitative parameters were analyzed via the Mann–Whitney U test. Essential indicators identified in a univariate analysis were utilized in a multivariate analysis to construct a logistic regression model and generate receiver operating characteristic curves (ROCs). Outcome prediction was assessed through the area under the curve (AUC) from the ROC analysis. The optimal diagnostic cutoff value was selected based on the highest Youden index point, with corresponding sensitivity and specificity calculations. A value of *p* < 0.05 was considered statistically significant.

## 3. Results

### 3.1. Clinical Information

A total of 106 TNBC patients were included in this study (age range, 22–71 years; mean age, 49.49 ± 9.970 years), with 30 (28.30%) in the NMHR group and 76 (71.70%) in the MHR group. Although the mean age of the NMHR group was slightly higher than that of the MHR group (49.90 ± 1.932 years vs. 49.33 ± 1.122 years), there was no statistically significant difference (*p* = 0.671). In terms of clinical staging, the T stage exhibited a significant correlation with pathological outcomes (*p* = 0.008). Specifically, the NMHR group had a substantially higher proportion of T4 tumors than the MHR group (53.33% vs. 23.68%). Conversely, the MHR group had a higher proportion of T2 tumors than the NMHR group (53.95% vs. 23.33%). However, there were no significant differences between the two groups in N staging and comprehensive clinical staging (*p* > 0.05). A summary of patient clinical information is provided in Table 1.

### 3.2. Conventional US Features

Twenty-five features were recorded and analyzed in this study (Table 2 and Appendix A Table A1). Notably, tumor size emerged as a significant factor influencing outcomes. The NMHR group exhibited a larger tumor size than the MHR group, with dmax (47.33 ± 3.538 mm vs. 34.39 ± 1.894 mm, *p* = 0.001) and volume (71,734.47 ± 17,862.349 mm^3^ vs. 23,359.86 ± 3343.984 mm^3^, *p* < 0.001) being notably different. While various changes in the tumor margin did not show individual associations with NAT efficacy, the NMHR group displayed a higher proportion of multiple changes in the margin (90.00% vs. 65.79%, *p* = 0.012) than the MHR group. Skin changes, either direct invasion or indirectly associated features, were significantly correlated with outcomes. Specifically, the MHR group had a lower proportion of skin changes (10.53% vs. 30.00%, *p* = 0.020), skin thickening (6.58% vs. 26.67%, *p* = 0.008), and skin invasion (9.21% vs. 26.67%, *p* = 0.030) than the NMHR group. Involvement of the retromammary space in the direct invasion layer was also associated with a poor treatment response (99.33% in NMHR vs. 69.74% in MHR, *p* = 0.010). After a specific analysis of various levels of lymph nodes, the abnormality of supraclavicular lymph nodes showed significant statistical significance, with 9 out of 30 (30.00%) in the NMHR group compared to 9 out of 76 (11.84%) in the MHR group (*p* = 0.025) displaying this abnormality.

No differences were observed in echo pattern, individual margin changes (angular, microlobulated, or spiculated), tumor orientation, calcifications, posterior acoustic effects, architectural distortion, duct changes, skin edema, invasion of various layers (subcutaneous fat, muscle, or nipple), other lymph node abnormalities (Levels I–III, subclavicular, internal mammary, or interpectoral), BI-RADS, or CDFI between the NMHR and MHR groups (*p* > 0.05).

### 3.3. SWE Parameters

The overall elastic parameters of the NMHR group were higher than those of the MHR group (Table 3). Notably, the NMHR group exhibited higher values for Emax (9.15 ± 0.290 m/s vs. 8.16 ± 0.239 m/s, *p* = 0.023), Emin (7.79 ± 0.411 m/s vs. 6.72 ± 0.259 m/s, *p* = 0.018), Emedian (8.57 ± 0.318 m/s vs. 7.41 ± 0.063 m/s, *p* = 0.022), and Emean (8.54 ± 0.314 m/s vs. 7.43 ± 0.249 m/s, *p* = 0.015). However, Estd and EIQR did not exhibit significance in the univariate analysis (*p* > 0.05).

### 3.4. Predictive Ability

Table 4 presents the model’s predictive ability determined through multiple factor logistic regression. US, SWE, and the combined model demonstrated predictive capabilities (*p* < 0.001). The combined model achieved the highest AUC value of 0.827, with an accuracy of 76.41%, a sensitivity of 90.47%, and a specificity of 55.81%. US followed closely with an AUC of 0.816, an accuracy of 81.13%, a sensitivity of 88.89%, and a specificity of 64.71%. SWE exhibited the lowest AUC of 0.734, with an accuracy of 71.70%, a sensitivity of 90.56%, and a specificity of 52.83% (Figure 3).

## 4. Discussion

Our study relied on the extensive ultrasound data recorded at baseline for TNBC patients undergoing neoadjuvant therapy (NAT). We meticulously compared baseline indicators to discern differences in NAT outcomes. Notably, our analysis unearthed a significant correlation between baseline ultrasound findings and the pathological response to NAT in TNBC.

Previous research has primarily relied on changes in tumor size to predict pathologic complete response (pCR) [26,27,28], often emphasizing midtreatment measurements as the key predictor [26]. However, for TNBC, which is associated with a mass in 70% to 86% of cases [29], the round mass of the tumor can develop into plaque-like areas, where the tumor volume reduces, but the diameter remains unchanged [30]. A recent study used US to simulate the probability of pCR at a critical point where the tumor volume decreased by ≥80% from baseline to two NAT cycles [28]. However, there is limited research on individual baseline conditions. According to the AJCC guidelines [21], T stage is determined based on the maximum diameter of the tumor. In our study, a higher T stage at baseline and larger ultrasound images with dmax and volume were more inclined toward NMHR after NAT. Combined with other US features, predictive conclusions can be better obtained.

Grayscale ultrasound imaging has emerged as a valuable tool not only in distinguishing benign and malignant tumors, but also in characterizing the molecular profile of malignancies. TNBC, in particular, often exhibits microlobular margins with clear boundaries and a notable absence of internal microcalcifications [8]. The formation of a tumor margin is due to the invasion and spread of tumor cells at the edge of BC to surrounding normal breast glands or adipose tissue, which causes an interstitial reaction, thus forming an irregular boundary zone. The edge band signs of BC multimode US are correlated with the tumor focus diameter, lymph node metastasis, and immunohistochemical markers, which have essential imaging significance for the accurate diagnosis of BC and the guidance of individualized treatment [31]. In addition, imaging studies on TNBC have shown that apparent elliptical or circular lesions are more likely to achieve pCR than diffuse or irregular lesions [32]. In this study, by refining the description of edge changes, the results show that in addition to blurring the tumor edge, whether there are other changes (angular, lobular, or spiculate) is an independent influencing factor for the efficacy of NAT.

Skin involvement was defined as skin thickening ≥ 2.5 mm in US or direct skin invasion. Nevertheless, only 25% of those with US-detected skin involvement have clinical skin involvement [33]. When skin thickening occurs in the breast, it indicates that the skin around the tumor is affected by lymphatic reflux disorders, increased blood supply, or venous congestion or that the tumor has directly invaded the skin [34], which may cause systemic spread [33]. Our research suggests that both direct skin invasion and indirect skin changes are associated with poor NAT efficacy.

The lymphatic system constitutes a pivotal route for breast cancer metastasis, mainly encompassing axillary, internal mammary, subclavicular, supraclavicular, and interpectoral lymph nodes, with the possibility of metastasis of 98.2%, 35.3%, 1.7%, 3.1%, and 0.7%, respectively [35]. In general, axillary lymph nodes (ALNs) and sentinel lymph nodes garner significant attention due to their prognostic value and correlation with NAT efficacy [36,37,38]. However, our TNBC-specific investigation did not establish a statistical relationship between Levels I, II, or III ALN abnormalities and NAT efficacy. It may be aimed at a single subtype of BC and cannot be used as an independent predictor. Nevertheless, a higher ALN positivity correlated with increased tumor malignancy, proliferation, and invasion, potentially leading to supraclavicular lymph node metastasis (SLNM) [39]. Regarding anatomical structure, supraclavicular lymph nodes are closely related to the accurate diagnosis and prognosis of BC. The prognosis of BC patients with ipsilateral supraclavicular lymph node metastasis (ISLNM), but no distant metastasis is poor [40,41]. This study emphasized the importance of ISLNM in the therapeutic effect through a comprehensive scan of breast-draining lymph nodes. Baseline imaging suggestive of ISLM, also known as synchronous ISLM, indicates poor NAT efficacy. Consistent with previous studies, pre-NAT imaging suggestive of synchronous ISLNM, followed by SLNM positivity after NAT, is a significant risk factor for survival after NAT [39]. In addition, the survival rate of ISCLM patients is influenced by the BC subtype. TNBC patients have the shortest survival period, and the prognosis of ISCLM caused by TNBC is the worst [42].

Conventional US combined with SWE can provide more imaging information for the early differential diagnosis of TNBC and improve the reliability of US diagnosis, with good clinical application prospects [8,43], reflecting the application value of the combined US model for TNBC. Research has suggested that more stiff tumors are less likely to experience clinical reactions and pCR [44]. However, some studies reported that Emax could be used to predict the response of invasive BC patients to NAC. Emean, Emax, and Emin before NAT are related to the response evaluation criteria in solid tumors (RECIST) index, but they cannot be used to predict the pCR of NAT [45]. Our study focused on the baseline SWE value of the TNBC, based on whether there was a significant pathological response rather than whether pCR status was achieved. The baseline stiffness of the MHR group was lower overall and could be used as a predictive indicator of NAT efficacy. In addition, more research has concentrated on the changes in the tumor treatment process, with predictive performance comparable to magnetic resonance imaging. For example, the changes in SWE after two or three cycles of NAT [16,18,30], as well as the degree of decrease in the Emean of lesions combined with changes in tumor diameter on conventional US [46], were all related to pCR. The focus was shifted to baseline SWE values as predictors of pathological response in our study, and it was observed that more favorable NAT efficacy corresponded to a lower baseline stiffness, particularly evident in the MHR group, underscoring the predictive potential of SWE. We also emphasize the clinical utility of combining conventional ultrasound (US) with shear-wave elastography (SWE). Although the addition of SWE did not significantly improve the predictive ability of US, as a quantitative imaging method, SWE can provide objective information that cannot be obtained from conventional US and initial clinical diagnoses, improving the credibility of judgment from another perspective [47,48]. At the same time, the acquisition of baseline SWE helps to monitor tumor changes in BC patients during NAT in the Chinese medical environment, making its promotion more meaningful.

It should be mentioned that the research shows that a higher Ki-67 (proliferation cell nuclear antigen) baseline value has a greater predictive significance for the pathological response of BC after NAT [48]. NAT can significantly reduce the Ki-67 index of BC, suggesting that Ki-67 can be used as an alternative index to predict the response of NAT, especially for TNBC [49]. However, our study might not have yielded compelling results on the expression of Ki-67 due to the influence of sample size; consequently, further in-depth research is warranted in this regard.

Our study has several limitations. The nature of the retrospective protocol and single-center research might have inevitable shortcomings. Despite the expertise of our sonographers, subjective factors in data collection remain unavoidable. Additionally, the diversity within TNBC, comprising seven subtypes, could not be individually analyzed due to sample size constraints. Therefore, our findings necessitate validation in larger, more heterogeneous patient populations.

## 5. Conclusions

In conclusion, our study highlights the significant relationship between baseline conventional US features and SWE parameters with the effectiveness of NAT for TNBC. The combination of these two models emerged as a noninvasive predictive tool, offering more reference for the efficacy of TNBC treatment, warranting early attention during the treatment’s initial stages.

## Figures and Tables

**Figure 1 diagnostics-13-03178-f001:**
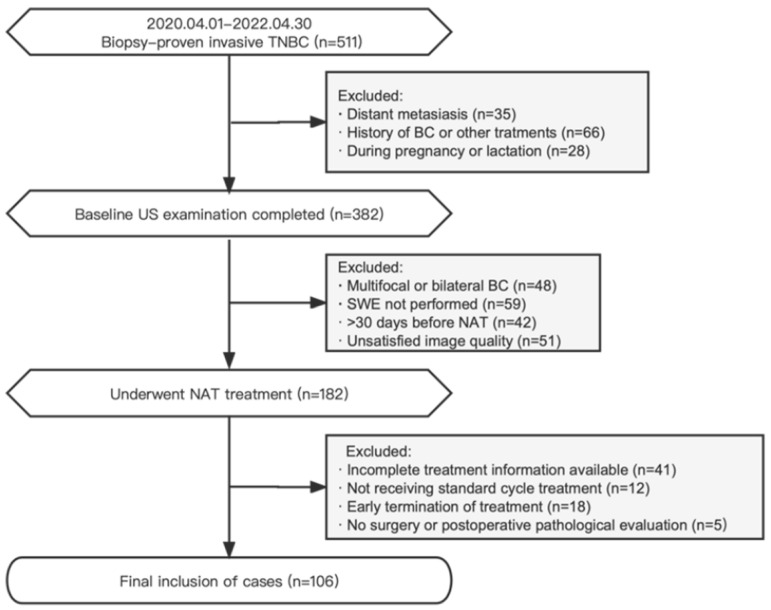
Flowchart of the research object in this study.

**Figure 2 diagnostics-13-03178-f002:**
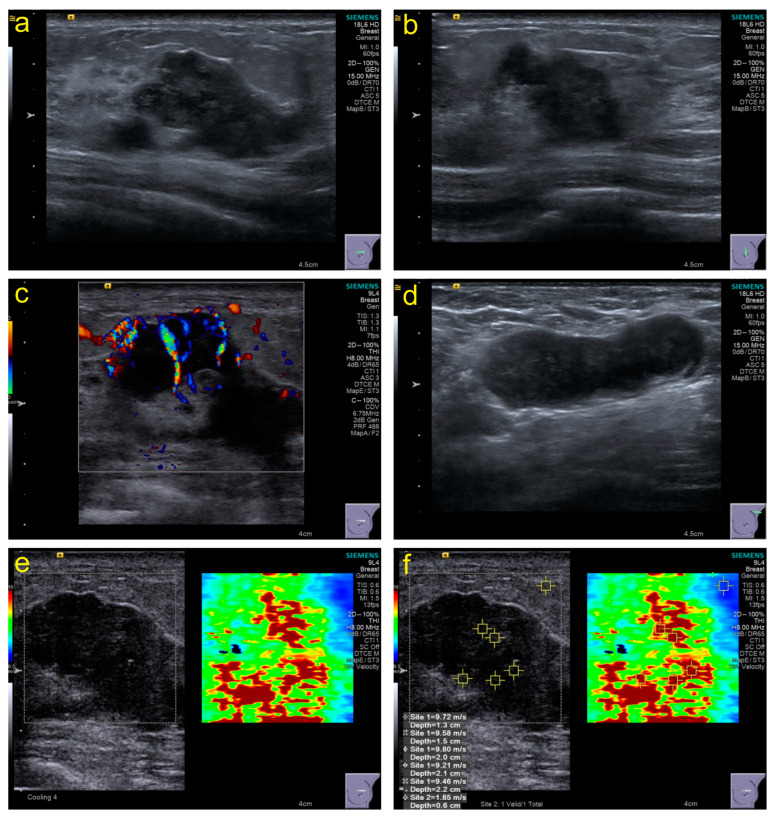
Examples of some necessary parts of sections in stored images and acquisition of SWE values. (**a**), the section with the maximum diameter; (**b**), the section perpendicular to the section with the maximum diameter; (**c**), the section with color Doppler flow imaging; (**d**), the section with the abnormal lymph node; (**e**,**f**), the section with shear-wave elastography and elasticity values acquisition.

**Figure 3 diagnostics-13-03178-f003:**
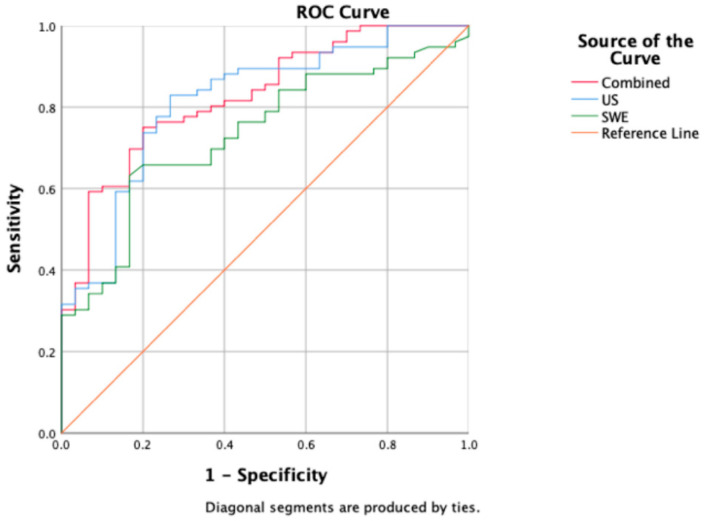
A comparison of ROC curves among US (blue), SWE (green), and combined model (red); corresponding AUC values are 0.816 (US), 0.734 (SWE), and 0.827 (combined model).

**Table 1 diagnostics-13-03178-t001:** The clinical data and histological characteristics of the patients included in this study.

Characteristic	Total (*n* = 106)	NMHR Group (*n* = 30)	MHR Group (*n* = 76)	*p*-Value
Age	49.49 ± 9.970	49.90 ± 1.932	49.33 ± 1.122	0.671
T stage				
1	7 (6.60)	1 (3.33)	6 (7.89)	0.008
2	48 (45.28)	7 (23.33)	41 (53.95)	
3	17 (16.04)	6 (20.00)	11 (14.47)	
4	34 (32.08)	16 (53.33)	18 (23.68)	
N stage				
0	14 (13.21)	4 (13.33)	10 (13.16)	1.000
+	92 (86.79)	26 (86.67)	66 (86.84)	
Clinical stage				
I	1 (0.94)	0 (0.00)	1 (1.32)	0.863
II	27 (25.47)	7 (23.33)	20 (26.32)	
III	78 (73.58)	23 (76.67)	55 (72.37)	
Ki-67%	49.86 ± 19.766	45.50 ± 3.175	51.58 ± 2.349	0.170
<14%	4 (3.77)	2 (6.67)	2 (2.63)	0.411
≥14%	102 (96.23)	28 (93.33)	74 (97.37)	

Abbreviations: NMHR, nonmajor histological response; MHR, major histological response; Ki-67: proliferation cell nuclear antigen. Continuous variables are mean ± standard deviation; *p* < 0.05, the difference is statistically significant.

**Table 2 diagnostics-13-03178-t002:** Key B-mode US features of the tumors included in this study.

Features		Total (*n* = 106)	NMHR Group (*n* = 30)	MHR Group (*n* = 76)	*p*-Value
Size	dmax	38.06 ± 18.241	47.33 ± 3.538	34.39 ± 1.894	0.001
	Volume	37,050.78 ± 61,074.369	71,734.47 ± 17,862.349	23,359.86 ± 3343.984	<0.001
Margin changes (angular, microlobulated, and/or spiculated)	(−)	29 (27.36)	3 (10.00)	26 (34.21)	0.012
	(+)	77 (72.64)	27 (90.00)	50 (65.79)	
Peripheral tissue	Skin changes (−)	89 (83.96)	21 (70.00)	68 (89.47)	0.020
	(+)	17 (16.04)	9 (30.00)	8 (10.53)	
	Skin thickening (−)	93 (87.74)	22 (73.33)	71 (93.42)	0.008
	(+)	13 (12.26)	8 (26.67)	5 (6.58)	
Invasion layers	Skin (−)	91 (85.85)	22 (73.33)	69 (90.79)	0.030
	(+)	15 (14.15)	8 (26.67)	7 (9.21)	
	Retromammary space (−)	25 (23.58)	2 (6.67)	23 (30.26)	0.010
	(+)	81 (76.42)	28 (93.33)	53 (69.74)	
Lymph node changes	Supraclavicular (−)	88 (83.02)	21 (70.00)	67 (88.16)	0.025
	(+)	18 (16.98)	9 (30.00)	9 (11.84)	

Abbreviations: US, conventional ultrasound; NMHR, nonmajor histological response; MHR, major histological response; dmax, maximum diameter of tumor; (−), feature absent; (+), feature present. Continuous variables are mean ± standard deviation; *p* < 0.05, the difference is statistically significant.

**Table 3 diagnostics-13-03178-t003:** The SWE parameters of the patients included in this study.

	Mean ± SD		Maximum		Minimum		*p*-Value
Group	NMHR Group (*n* = 30)	MHR Group (*n* = 76)	NMHR Group (*n* = 30)	MHR Group (*n* = 76)	NMHR Group (*n* = 30)	MHR Group *(n* = 76)	
Emax	9.15 ± 0.290	8.16 ± 0.239	10.00	10.00	3.07	2.55	0.023
Emin	7.79 ± 0.411	6.72 ± 0.259	10.00	10.00	1.08	2.21	0.018
Emedian	8.57 ± 0.318	7.41 ± 0.063	10.00	10.00	2.68	2.40	0.022
Emean	8.54 ± 0.314	7.43 ± 0.249	10.00	10.00	2.63	2.40	0.015
Estd	0.53 ± 0.075	0.63 ± 0.056	1.39	2.10	0.00	0.00	0.366
EIQR	0.91 ± 0.140	1.07 ± 0.109	2.72	5.13	0.00	0.00	0.422

Abbreviations: NMHR, nonmajor histological response; MHR, major histological response; Emax/Emin/Emedian/Emean, maximum/minimum/median/mean elasticity; Estd/EIQR, standard deviation/interquartile range of elasticity. Continuous variables are mean ± standard deviation; *p* < 0.05, the difference is statistically significant.

**Table 4 diagnostics-13-03178-t004:** AUC, accuracy, sensitivity, and specificity of US, SWE, and combined regression models.

Mode	AUC	95% CI	Accuracy (%)	Sensitivity (%)	Specificity (%)	*p*-Value
US	0.816	0.727–0.905	81.13	88.89	64.71	<0.001
SWE	0.734	0.635–0.832	71.70	90.56	52.83	<0.001
Combined	0.827	0.745–0.909	76.41	90.47	55.81	<0.001

Abbreviations: AUC, area under the curve; 95% CI, 95% confidence interval; US: conventional ultrasound; SWE, shear-wave elastography. *p* < 0.05, the difference is statistically significant.

## Data Availability

The original contributions presented in this study are included in the article. Further inquiries can be directed to the corresponding author.

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
