# Peer review of "Correlation between Baseline Conventional Ultrasounds, Shear-Wave Elastography Indicators, and Neoadjuvant Therapy Efficacy in Triple-Negative Breast Cancer"

_diagnostics, 2023, doi:10.3390/diagnostics13203178_

Round 1

Reviewer 1 Report

This is a very interesting study that adds to the body of literature into non-invasive methods of predicting pathological response to neoadjuvant chemotherapy in triple negative breast cancer. The results show combined US and SWE have a relatively reasonable predictive value. 
As a breast surgeon, I am familiar with US, but not so much with SWE. The introduction of the article does not provide enough info about this test, or its routine use and how it works. The introduction needs to change to address this. 
How does this predictive tool compare to other conventional radiology methods, like mammogram, MRI and CT? 
The mean age of this cohort is young, 49 years, do we know how many were BRCA carriers and how does that impact the test? 
The study included a relatively small sample size which may not allow subgroup analysis, however, it would interesting to see how well did the test do in those who had non major response as these are the target group that might benefit from treatment modification. The specificity of this combined test is about 76.41 and sensitivity is 90.47, this is relatively low for a test to determine response to NAT, so I do not agree with the authors conclusion and as per these results alone, it would hardly be used in practice for these patients, especially with no discussion regarding the learning curve and how to standardise SWE which is not used routinely internationally.
The first sentence in the abstract needs to be reviewed from an English language/grammar point of view. The results table are good and informative but quite long and difficult to follow across pages. 

Good

Reviewer 2 Report

Thank you for letting me review this paper on ultrasound and elastography of triple negative breast cancer tumours before neoadjuvant chemotherapy. The aim was to correlate baseline findings with tumour response.

It is difficult to follow the paper and it need a thorough check of the English language. Maybe it is due to this that I do not understand how the multiple-factor analysis was done. What factors were included?

In the Discussion, paragraph 2, you state that regardless of size etc IT can be correlated with the final efficay of NAT. Please, specify it. If you mean ultrasound you have to explain in more detail how you came to this conclusion.

Explain all abbreviations in all tables. Explain CDFI in the text.

First paragraph in Introduction. HER2 negativity is part of the definition of triple negativity.

Second sentence in Results, there is no statistically significant difference in age between response groups. You are refering to difference in respons.

Table 2, in the heading you refer to US features of the patients, I would prefer "tumours".

Extent the figure legend for figure three.

In table 1 you can show the results for N0 versus N+ patients instead.

See above

Round 2

Reviewer 2 Report

Results. Line 152-154. "The average ages in both groups were comparable (49.90±1.932 years vs. 49.33±1.122 years), and there was no statistically significant difference in their response to NAT (P=0.671)." This is not what you mean. Age and response is not presented. You present response and age. Re-write.

Author Response

Comment: Results. Line 152-154. "The average ages in both groups were comparable (49.90±1.932 years vs. 49.33±1.122 years), and there was no statistically significant difference in their response to NAT (P=0.671)." This is not what you mean. Age and response is not presented. You present response and age. Re-write.

Reply: Sincerely thank you for reviewing our manuscript again, and we hope that we have understood your comments correctly. We have re-written this part according to your comment, as detailed in line 151-155.